# Single and Combined Effects of Preferred Music and Endpoint Knowledge on Jump Performance in Basketball Players

**DOI:** 10.3390/sports11050105

**Published:** 2023-05-15

**Authors:** Nidhal Jebabli, Mariem Khlifi, Nejmeddine Ouerghi, Manar Boujabli, Anissa Bouassida, Abderraouf Ben Abderrahman, Roland van den Tillaar

**Affiliations:** 1Research Unit: Sports Science, Health and Movement, UR22JS01, High Institute of Sport and Physical Education of Kef, University of Jendouba, Le Kef 7001, Tunisia; 2Faculty of Medicine of Tunis, Rabta Hospital, LR99ES11, University of Tunis El Manar, Tunis 2092, Tunisia; 3High Institute of Sport and Physical Education of Ksar Said, Manouba University, Tunis 2010, Tunisia; 4Tunisian Research Laboratory “Sports Performance Optimization”, National Center of Medicine and Science in Sports (CNMSS) LR09SEP01, Tunis 1003, Tunisia; 5Department of Sports Sciences and Physical Education, Nord University, 8026 Levanger, Norway

**Keywords:** fast tempo music, prior knowledge endpoint type, repeated countermovement jumps, psychophysiological responses

## Abstract

Both music and endpoint knowledge of exercise have been shown to independently influence exercise performance. However, whether these factors work as synergists or counteract one another during exercise is unknown. The purpose of this study was to determine the single and combined effect of listening to preferred music and types of endpoint knowledge on repeated countermovement jump (CMJ) test performance. Twenty-four (n = 24) current or previously competitive basketball players underwent CMJ testing under the following endpoint knowledge conditions: (1) unknown/no knowledge, (2) knowledge of the number of jumps, and (3) knowledge of exercise duration. For each of these, participants listened to either their preferred music or no music during the duration of testing. For the exercise portion, participants completed repeated CMJs where participants were encouraged to jump as high as possible with jump height, contact time, and flight time as outcomes. Rate of perceived exertion (RPE) and feeling scale were measured before and after exercise. The results showed that, regardless of knowledge type, preferred music resulted in a significant decrease in both contact time and flight time (F ≥ 10.4, *p* ≤ 0.004, and η_p_^2^ ≥ 0.35), and a significant improvement of jump height (F = 11.36, *p* = 0.001, and η_p_^2^ = 0.09) and feeling scale ratings (F = 36.9, *p* < 0.001, and η_p_^2^ = 0.66) compared to no-music condition, while RPE was not significantly affected. Regardless of the presence of music, knowledge of the number of jumps and duration resulted in lower contact time (*p* < 0.001, 0.9 < d < 1.56) versus unknown condition during CMJs. Moreover, a significant decrease in RPE values was found during prior endpoint knowledge of number (*p* = 0.005; d = 0.72) and duration (*p* = 0.045; d = 0.63) compared to unknown condition. However, feeling scale ratings were not significantly affected. Moreover, no interactions with significance findings were found for any parameters. Overall, data suggest that listening to music and endpoint knowledge alter exercise responses in basketball players, but they do not interact with one another.

## 1. Introduction

Fatigue is generally considered as a negative consequence of acute physical exercise, under the influence of continuous control of central and peripheral factors [1,2]. Regulation of “pace”, called pacing strategy, reflects the conscious and/or subconscious variation of the workload during exercise to limit the premature onset of fatigue factors in attempts to increase physical performance [3,4]. Music has been well-established to enhance physical performance largely through synchronization and pacing of movement to musical tempo [5,6]. Furthermore, music has been shown to modulate various psychophysiological processes during exercise, which may manifest concurrently or independently of performance enhancement [7]. However, it remains unclear how the efficacy of listening to music during exercise is influenced by other peripheral factors.

Awareness or anticipation of when physical effort will cease, also termed “teleoanticipation” or “endpoint knowledge”, has been established as a factor that alters responses to vigorous exercise [8]. In conditions where endpoint knowledge is lacking, energy production and rate of perceived exertion (RPE) have been shown to be lower, which suggests possible increases in work economy compared to a condition in which the endpoint is known [9]. Reductions in RPE likely depend on the ability to reduce the consumption of the energy reserve in case the duration exceeds the ability of a faster pace. Coquart et al. [10] observed that athletes perceived triangular exercise on a cycle ergometer as being easier during a test with an unknown endpoint (open loop) compared to the test with a running distance or a known duration (closed loop). In addition, previous studies have shown that a planned pacing strategy in which the prior knowledge of the endpoint of the intense exercise, prepared to be applied by the athlete, enhances performance without fully depleting before the end of the task leading to it [11,12,13]. Although the physiological determinants are debated, central regulatory control (i.e., central governor theory) has been implicated to mediate pacing and exercise intensity by controlling metabolic resources and motor output [14]. Others have also suggested that central regulatory control might be influenced by other external factors, such as psychological aids (i.e., music, verbal encouragement, etc.) that can optimize pacing strategy and improve athletic performance. While most evidence suggests changes in behavior during exercise with prior endpoint knowledge, how other psychological stimuli influence this remains unknown.

Listening to music has been shown to impart ergogenic effects in a wide range of exercise modalities [15]. As a psychological aid during exercise, listening to music has been shown to improve mood/affect, enjoyment, motivation, and RPE [16]. Furthermore, inherent characteristics of music, such as musical tempo, have been shown to alter pacing strategy and improve exercise efficiency in aerobic [17,18] and anaerobic exercise [19,20]. However, conflicting reports exist with some studies reporting little to no effects of music on jump height during squat jumps [21].

For team sport, it is well documented that preferred music can be beneficial, as an ergogenic aid, to improve specific abilities in basketball [22,23,24,25], considering that basketball is a team sport where jumps have a key importance to score in this sport [26]. Indeed, Ben Abdelkrim et al. [26] report that professional basketball players perform an average of 44 ± 6.7 jumps during a basketball game. However, to our knowledge, no study has evaluated the effect of music on the physical performance of repeated jumps exercise. From an overview of those previous literature, it remains unclear whether listening to music has a beneficial effect on performance of repeated jumps by basketball players.

While listening to music and endpoint knowledge have been separately studied in the context of performance, little is known as to how they interact or if concomitant occurrence results in synergistic responses to explosive exercise [27]. Both endpoint knowledge and listening to music may affect the cognitive and/or psychological determinants of exercise performance [11,28], but little is known regarding this in the context of explosive exercise. Since having endpoint knowledge tends to decrease performance and listening to music increases exercise ability, it is plausible that possessing endpoint knowledge may interfere with music-induced dissociation and counteract benefits during repeated exercise. To our knowledge, there is no study that has shown that the prior knowledge of the endpoint takes place during high repeated vertical jump exercise, especially with music. Thus, the purpose of this study was to determine the single and combined effects of listening to preferred music and types of endpoint knowledge on repeated CMJ test performance. Based on previous findings [11,19], we hypothesized that listening to preferred music and prior knowledge of duration endpoint during repeated CMJ until exhaustion could separately improve physical performance compared with the condition of no music or another knowledge of endpoint type.

## 2. Materials and Methods

### 2.1. Experimental Approach to the Problem

To assess the effect of combined listening to preferred music and prior knowledge of endpoint on physical and psychophysiological variables during a self-paced jumping test, this study adopted a randomized controlled design. Participants performed eight sessions during three weeks, including two days of familiarization and six experimental sessions. Each test session had 48–72 h of recovery. All the sessions were carried out at the same time of day (afternoon at 15–17 h), with constant environment conditions (temperature = 20–22 °C, relative humidity = 15–40%), and at an altitude of ~900 m.

During the experimental protocol, each participant completed six separate sessions. The first session consisted of a CMJ test with CMJ undertaken to exhaustion. In the following second to sixth sessions, in random order, participants completed the same repeated CMJ where they were informed, repeating the CMJ test with the same time as participants who completed the first session (repeating CMJ to exhaustion). Then they repeated the CMJ test with the same frequency (number) of jumps that participant completed in the first session (repeating CMJ to exhaustion). These sessions were also repeated with the same design with listening to preferred music (Figure 1). Randomization was carried out with regards to the experimental conditions using a freeware online tool (https://www.randomlists.com/team-generator, accessed on 26 March 2023).

With the purpose of reducing bias, the subjects were not informed of the hypotheses and expectations based on prior research. Also, the statistician was blinded to the experimental purpose and intervention allocation.

The present study adopts a research design of Ato et al. [29].

### 2.2. Participants

To determine adequate sample size, an a priori power analysis was calculated using G*Power (version 3.1.9.2, University of Kiel, Kiel, Germany) using the f test family (ANOVA repeated measures, two fixed factors, f = 0.6, α = 0.05) of a similar study of Jebabli et al. [17]. The analysis revealed that a total sample size of n = 24 would be sufficient to find significant and medium-sized effects of condition (effect size) with an actual power of 0.90. Thus, 24 healthy male sport-science students (age 22.09 ± 1.16 years; weight 72.73 ± 9.31 kg; height 1.79 ± 0.13 m; and BMI 22.61 ± 2.20 kg·m^−2^) volunteered to participate in this study.

Participants were active in basketball (6 h·wk^−1^) and with previous competitive experience (3.5 ± 1 years). No neuromuscular disorders or specific musculoskeletal injuries to the ankles were previously experienced by participants. Also, participants were not allowed to drink any beverage containing an ergogenic product (e.g., caffeine, vitamins, protein, etc.) or alcohol during the experimental protocol. All participants signed informed consent forms outlining commitment, benefits, potential risks, and the study’s procedures, which were in accordance with the local ethics committee of the Higher Institute of Sport and Physical Education of Kef (EC-UR22JS01-010-2018), Jandouba, Tunisia, and the protocol was carried out in accordance with the latest version of the Declaration of Helsinki.

### 2.3. Procedures

#### 2.3.1. Familiarization Session

Two weeks before the start of the experimental trials, two familiarization sessions separated by one week were performed, in order to restrict learning effects during the experiment and decrease the risk of injury. In each familiarization session, participants initially performed a standardized warm-up (10 min). After 3 min of passive rest, participants performed the CMJ and then repeated the CMJ until exhaustion. Also, participants self-selected their preferred music which they felt more inclined to choose for the repeated CMJ. Moreover, anthropometrics measurements were taken. In addition, all instruments and equipment were explained and practiced, such as the use of the same wireless Bluetooth headphones, without listening to music, during the tests.

#### 2.3.2. Experimental Protocol

The participants completed six experimental sessions:

##### Session 1. Repeated CMJ to Exhaustion

Participants were required to do the repeated CMJ test until exhaustion (unknown condition). They were told that jumps heights would be maximal (all out) and the first jumps would be >95% of the previous performance of CMJ (familiarization session).

Participants were kept unknowing of the duration, height, and jump number of the test when they did not have prior knowledge of test endpoint.

##### Sessions 2, 3. Prior Knowledge Duration or Number

In random order, participants were asked to jump the same time or the same number of jumps completed in Session 1.

The investigator used Microgate software (Optojump software, version 1.10.50) to track the number or duration of jumps in order to stop the participant when he or she completed the test.

##### Sessions 4–6

The players repeated the first to third sessions listening to their preferred music.

### 2.4. Music Preference

During the CMJ test and familiarization sessions, preferred music was preselected for each participant: participants were instructed to preselect their preferred music to be played during the repeated CMJ test. Using the “Edjing Mix” application (version 6.45.00, Android, France), we observed that all participants chose high-tempo music (>140 bpm). Music was played from an mp3 player through the same wireless Bluetooth headphones (Jabra Elite 65t, Ballerup, Denmark) during all sessions, and the music was turned off immediately after the test. Volume was standardized to the same level (moderate volume, 70 dB) for all participants in all sessions. During the no-music condition, wireless Bluetooth headphones were worn but no music was played.

### 2.5. Measurements

#### 2.5.1. Countermovement Jumps Exercise

From an upright standing position, the participants performed many fast vertical jumps with downward eccentric and upward concentric actions. They were informed to jump as high as possible with both hands on the hips during the action to eliminate any influence of arm swing. Before the repeated CMJ started, participants completed a standardized 10 min warm-up that included 4 min of light jogging, lateral displacements, dynamic stretching, and jumping, followed by 3 min of passive recovery. The Optojump-next device (Bolzano, Italy) connected to a laptop via USB and the Microgate software (Optojump software, version 1.10.50) were used. Jump duration(s), number of jumps, jump height (cm), contact time(s), and flight time(s) were recorded for analysis.

Pilot data from 24 participants collected on two different days were used to determine the reproducibility of the test (ICC 0.975; 95% CI: 0.899–0.99).

#### 2.5.2. Ratings of Perceived Exertion

The ratings of perceived exertion (RPE) leg (6–20 Borg scale; [30]) was used to measure the physical perceptions of exertion for legs just after repeated CMJ test. Ratings of perceived exertion had assessed verbally after repeated CMJ using the 15-point version of the RPE scale with ratio properties, which ranges were from 6 (no exercise) to 20 (maximum exercise).

#### 2.5.3. Feeling State

Feeling state (FS) was measured to assess the participant’s mood after the repeated CMJ test. FS was an 11-point single item scale ranging from +5 (very good) to −5 (very bad) with a midpoint of 0 (neutral) [31]. Participants received standard instructions regarding the use of the FS, after repeated CMJ test, according to Hardy and Rejeski [31], as it is quite common that mood changes during exercise. Some individuals find exercise pleasurable, whereas others do not. Therefore, this scale was developed to measure such responses. Previous studies recommended this scale to measure affective response after exercise [18,19,20,21].

### 2.6. Statistical Analysis

Data were expressed as means and standard deviations (SD). Normality of data was assessed and confirmed using the Shapiro–Wilk test. The reliability of the CMJ and repeated CMJ tests was assessed by the intra-class correlation coefficient (ICC) and 95% confidence interval (CI). The data of repeated CMJ (average height, mean flight time, and mean contact time), RPE, and feeling scale were examined using univariate ANOVA, with two fixed factors (music condition, and endpoint knowledge type). When significant differences were found, a Holm–Bonferroni probability adjustment post hoc test was used to determine the source of those differences. Where the sphericity assumption was violated, the Greenhouse–Geisser adjustments of the *p*-values were reported. The effect size was evaluated with η_p_^2^ (partial eta squared) where 0.01 < η_p_^2^ < 0.06 represents a small effect, 0.06 < η_p_^2^ < 0.14 a medium effect, and a large effect when η_p_^2^ > 0.14. Cohen’s d (d) [32] was calculated to quantify meaningful differences in the data with demarcations of trivial (<0.2), small (0.2–0.59), medium (0.60–1.19), large (1.2–1.99), and very large (≥2.0).

Besides, significance was set at *p* < 0.05. Statistical analysis was performed using Statistical Package for the Social Sciences version 16.0 software (SPSS Inc., Chicago, IL, USA).

## 3. Results

### 3.1. Physical Performance

Total number of jumps was not significantly affected by music (F = 2.05, *p* = 0.168, and η_p_^2^ = 0.10). However, a significant effect of prior knowledge was found (F = 6.0, *p* = 0.005, and η_p_^2^ = 0.24). Furthermore, no significant music* type and endpoint knowledge interaction (F = 0.37, *p* = 0.69, and η_p_^2^ = 0.01) was found. Post hoc comparisons revealed significantly more jumps when jumping with preferred music when the knowledge of number of jumps was known compared with no music. Furthermore, when jumping with music, prior knowledge of duration resulted in significantly more jumps (Figure 2).

Contact and flight times were significantly affected by music (F ≥ 10.4, *p* ≤ 0.004, and η_p_^2^ ≥ 0.35). Also, jump height and contact times were significantly affected by prior knowledge type (F ≥ 4.1, *p* ≤ 0.024, and η_p_^2^ ≥ 0.18), while no significant prior knowledge effect on flight time and interaction effects for any of these three parameters were found (F ≤ 2.9, *p* ≥ 0.096, and η_p_^2^ ≤ 0.10). Post hoc comparisons revealed that jumping height was higher, contact time shorter, and flight time longer when using preferred music in each prior knowledge condition compared with no music. Furthermore, in the unknown knowledge condition, jump height was significantly lower than with prior knowledge about number of jumps, and contact time was significantly longer in the unknown knowledge condition than with the other two knowledge conditions (Figure 3).

### 3.2. RPE and Feeling Scale

A significant main effect of endpoint knowledge was found for RPE (F = 6.7, *p* = 0.003, and η_p_^2^ = 0.26), but not on the feeling scale (F = 0.32, *p* = 0.73, and η_p_^2^ = 0.02) (Figure 4). The feeling scale was significantly affected by music (F = 36.9, *p* < 0.001, and η_p_^2^ = 0.66), while RPE was not significantly affected, but had a large effect on size (F = 4.13, *p* = 0.056, and η_p_^2^ = 0.18). No significant interaction effects were found for these two parameters (F ≤ 3.2, *p* ≥ 0.077, and η_p_^2^ ≤ 0.14). Post hoc comparisons revealed that the feeling scale was significantly more positive when performing jumps with preferred music in each condition, while RPE was significantly higher in the unknown knowledge condition with other conditions, and that RPE was higher with preferred music in the unknown and known number situations with the no-music condition.

## 4. Discussion

The aim of this study was to investigate the single and combined effects of preferred music and endpoint knowledge on jump performance in basketball players. The main finding of this study was that preferred music and prior knowledge application, during repeated CMJ, separately affected performance jump height, contact time, flight time (only with music), and jump number (only with knowledge condition). Furthermore, preferred music provided a positive enhancement on the feeling scale. Finally, prior endpoint knowledge conditions during listening to preferred music decreased significantly RPE values compared to unknown condition. These results did not support the experimental hypothesis.

In the current study, the significant main effect of music indicated that there was a significant increase in average jump height and flight time, accompanied by a reduction of contact time, during preferred music sessions compared to no music. However, research has shown that positive effects of music are more likely to be felt during endurance activity than during short-term exercise [33]. Also, the improvements in jump height average and flight time while listening to preferred music is in contrast to previous literature as Ghazel et al. [19], who found no changes in CMJ performance using fast music (140 BPM) in female handball players. The difference in music preferences (preferred, not preferred), participant fitness level (trained, untrained), and gender (male, female) could be the cause of the discrepancies between these findings and that of the present study. Additionally, discrepancies in results can be partially explained by the rhythmic character of music and its pacing effect. While maximal nature of repeated CMJ (all out), pacing to music may not only be non-beneficial, but could also undermine the benefits of preferred music [34].

The present study demonstrated that prior knowledge of number and prior knowledge of the duration leads to a significant decrease in contact time with or without listening to preferred music, and an increase in mean jump height and jump number compared to the unknown condition. However, no significant differences were observed for flight time. To our knowledge, no similar study has been conducted, and the present research offers new insights into this context. However, the specificity of basketball player training may be a factor contributing to greater jump performance when he or she knows the endpoint in advance. This is because the rules of basketball require players to complete each offense or defense before 24 s (prior knowledge time), and they are more trained on specific tasks, during training, with prior knowledge of the endpoint.

In physiological order, it seems that prior knowledge of duration or number favors more effort regulation [13]. These central nervous system-controlled mechanisms have some influence on physical performance, even during intense physical activity [35,36]. Probably through these regulations, prior knowledge of the duration seems to reduce the contact time during exercise and RPE. Further supporting this, Hanson and Buckworth [37] showed that knowledge of exercise duration improves running performance during aerobic exercise compared to the unknown condition. Similarly, Highton et al. [38] examined the effect of different type of endpoint knowledge (known, disappointment, and unknown) on variation in pace and physical performance in simulated rugby league matches. They noted that knowing the duration of the exercise favors a significant improvement in the pace and physical performance of the players compared to the other conditions. In the same context, previous studies have shown that when the number of repetitions of the trials during a physical exercise is known, the athletes apply more physical effort compared to the unknown condition [12]. In agreement with the present results, these studies affirmed that the pre-knowledge of endpoint before each physical exercise presents the best information to have an optimal performance for the exercises of aerobic and anaerobic origin.

The results of the current study showed that only preferred music promotes a significant improvement in the mood state of participants during repeated CMJ compared to no music.

To our knowledge, no study has investigated the effect of favorite music on mood state during exercises based on vertical jumps. However, the results of the present study are in agreement with the study of Aloui et al. [39], who reported that listening to music during warm-up positively influences variation in physical performance for short-term exercise, cognitive anxiety, self-confidence, and mood.

Our results showed that prior knowledge number and prior knowledge duration during repeated CMJ with preferred music condition promoted a significant decrease in RPE value compared to unknown state. To our knowledge, no study has investigated the effect of favorite music and prior knowledge endpoint on RPE during repeated CMJ test. However, the present results contrast with those of Billaut et al. [13] who observed that the different types of knowledge endpoint (known, unknown, and disappointed) during repeated sprint exercises (RSE) did not have a significant effect on the RPE value.

These differences could be explained by the difference in the types of exercises (CMJ repeated vs. repeated sprint), the type of knowledge of endpoint (known, disappointment, unknown vs. unknown, prior knowledge duration, and prior knowledge number) and the duration of the exercise (30 s vs. 6 s). The variability of the RPE could also be affected by the concept of tele-anticipation (prior knowledge of the endpoint), whose feedback could reduce the consumption of metabolic sources during physical exercise. This sensation of fatigue figured by the decrease in the RPE compared to the unknown condition. In addition, the significant reduction in RPE when assessing preferred music could be explained by the fact that listening to music plays a role in distracting sensory and physical cues during exercise [40].

## 5. Conclusions

In conclusion, the findings of the present study indicate that listening to music and prior knowledge number or duration positively affect repeated CMJ, RPE, and feeling scale responses in basketball players, but do not interact with each other. Based on the results of the present study, we encourage basketball conditioning experts to use preferred music and prior knowledge as two processes to enhance physical and psychophysiological responses during the repeated CMJ test. Future research is needed to better understand their effects on other technical, psychological, and physiological parameters in male and female basketball players.

## Figures and Tables

**Figure 1 sports-11-00105-f001:**
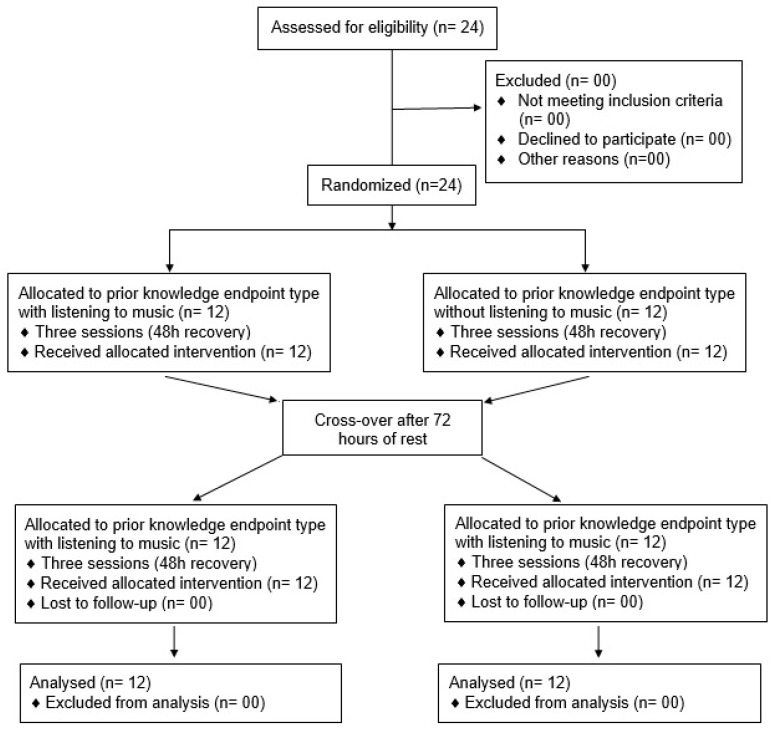
Flow diagram.

**Figure 2 sports-11-00105-f002:**
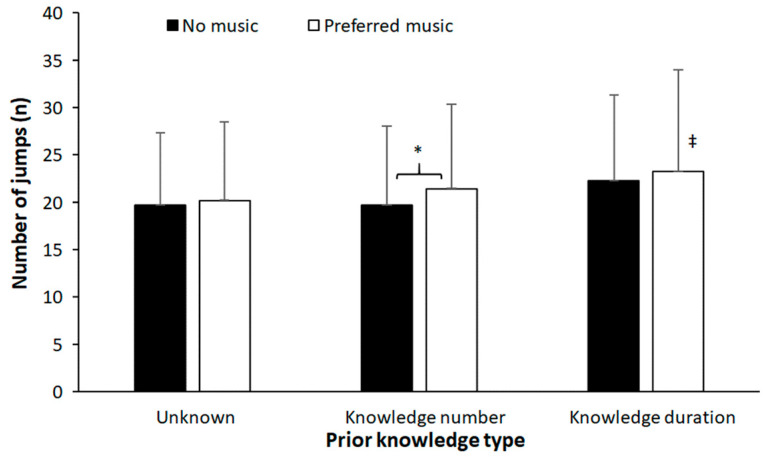
Single and combined effects of preferred music and endpoint knowledge on number of jumps. * indicates a significant effect of music in this knowledge type. ‡ indicates a significant difference with the other knowledge types for this condition (*p* < 0.05).

**Figure 3 sports-11-00105-f003:**
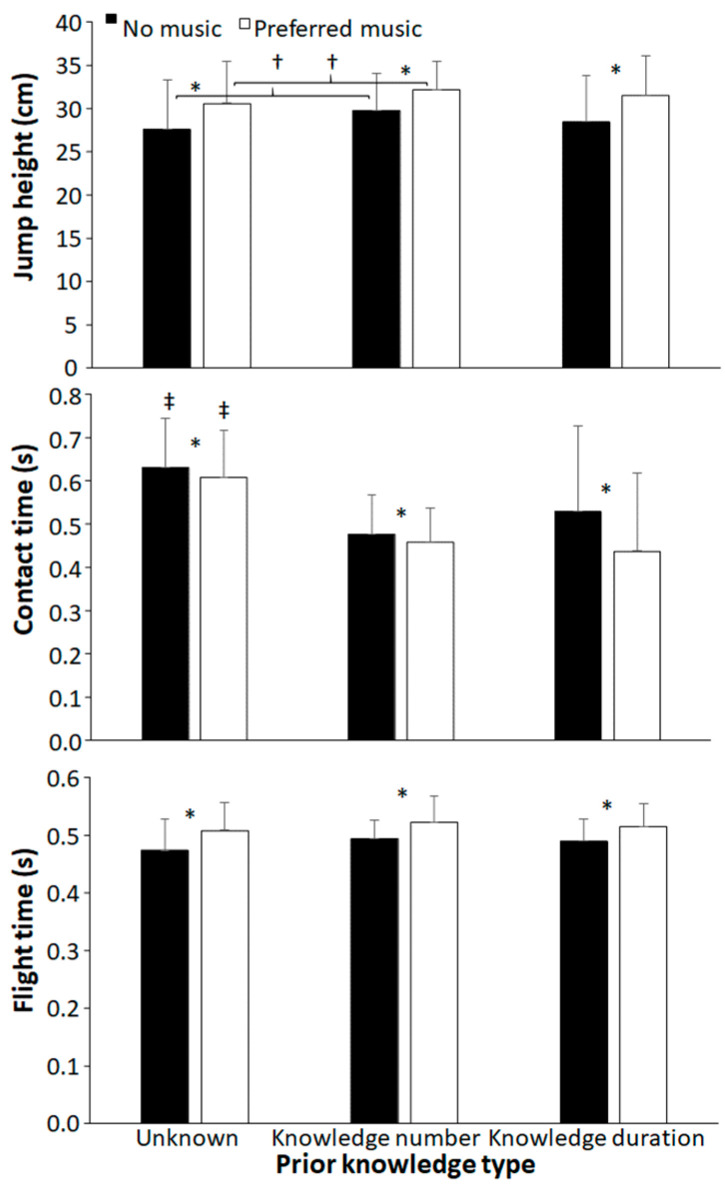
Single and combined effects of preferred music and endpoint knowledge on average jump height, contact, and flight time. * indicates a significant effect of music in this condition. ‡ indicates a significant difference with the other knowledge types for this condition. † indicates a significant difference between these two knowledge types for this condition (*p* < 0.05).

**Figure 4 sports-11-00105-f004:**
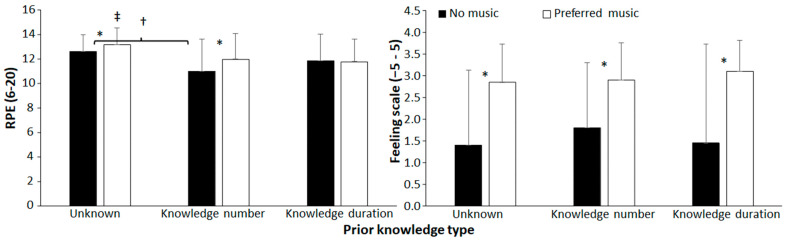
Single and combined effects of preferred music and endpoint knowledge on average RPE and feeling scale. * indicates a significant effect of music in this condition. ‡ indicates a significant difference with the other knowledge types for this condition. † indicates a significant difference between these two knowledge types for this condition (*p* < 0.05).

## Data Availability

Not applicable.

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
