# Peer review of "Single and Combined Effects of Preferred Music and Endpoint Knowledge on Jump Performance in Basketball Players"

_sports, 2023, doi:10.3390/sports11050105_

Round 1
Reviewer 1 Report
Please provide as much clarification of the methodology as possible and complete the discussion based on your assumptions for this study.

Author Response
Se attached file

Reviewer 2 Report
- the authors aimed to determine the single and 16 combined effect of listening to preferred music and types of endpoint knowledge on repeated coun- 17 termovement jump test performance. some minor comments for review:
- line 80: cite articles.
- line 100: and one session of...?
- report all input parameters and report studies that support the use of medium effect size.
- report the moment application f the RPE and FS scale. This needs to be disclosed since reported variability could occur before, during, and after (time?) test.
- In addition, report how the authors anchored endpoints of these measures with the participants for clarity.
- figure 3 is missing.
- limitations should be disclosed.
Author Response
Thank you for giving us the opportunity to submit a revised draft of the manuscript “Single and Combined Effects of Preferred Music and Endpoint Knowledge on Jump Performance in Basketball Players.” for publication in Sports MDPI. We appreciate the time and the effort that you dedicated to providing feedback on our manuscript and we feel grateful for the insightful comments and valuable suggestions. We have incorporated most of the suggestions made by you and the reviewers and carefully considered your concerns. All changes for reviewer 2 are highlighted in green throughout the revised manuscript. Please see below, in red, for a point-by-point response to the reviewers’ comments and concerns.
Reviewer 2
- the authors aimed to determine the single and combined effect of listening to preferred music and types of endpoint knowledge on repeated countermovement jump test performance. some minor comments for review:
- line 80: cite articles.
Done as you suggested.
- line 100: and one session of...?
Statement has been changed (line 105-106): “Participants performed eight sessions during 3 weeks, including 2 days of familiarization, and six experimental sessions”
- report all input parameters and report studies that support the use of medium effect size.
The following statement has been changed and study support has been added (line 126-131): “To determine adequate sample size, an a priori power analysis was calculated using G*Power (Version 3.1.9.2, University of Kiel, Kiel, Germany) using the f test family (ANOVA repeated measures, two fixed factors, f = 0.6, α= 0.05) of a similar study of Jebabli et al. [17]. The analysis revealed that a total sample size of N = 24 would be sufficient to find significant and medium-sized effects of condition (effect size) with an actual power of 0.90.”
- report the moment application f the RPE and FS scale. This needs to be disclosed since reported variability could occur before, during, and after (time?) test.
The RPE scale was presented just after repeated CMJ test, immediately followed by the FS scale. The following statement has been added (line 206-207):” The RPE scale was presented first, immediately followed by the FS”
- In addition, report how the authors anchored endpoints of these measures with the participants for clarity.
Done as you suggested.
- figure 3 is missing.
Sorry. Figure 3 (actually figure 4) has been added (line 271).
- limitations should be disclosed.
We thank the reviewer for his/her comments. All conclusion section has been modified
Reviewer 3 Report
The authors present a good research paper.
- The relevance of the topic: Good.
- Introduction: Can be improved.
- Methodology: Can be improved.
- Results: Good.
- Discussion: Good.
However, ACCEPT AFTER MINOR REVISION. In general, the paper follows an adequate structure and correct scientific support and can be published considering some limitations. The study is interesting in the field of Basketball. However, there are a series of limitations that should be considered.
In the first place, carry out a review of the existing literature related to the subject, being essential to inquire into the MPDI – Sports journal itself, since there are papers related to its manuscript that can help to improve it. Therefore, include those references, if any, especially from the last five years. In addition, recommend reading some papers related to the topic of Basketball and Music:
Chen, C. C., Chen, Y., Tang, L. C., & Chieng, W. H. (2022). Effects of interactive music tempo with heart rate feedback on physio-psychological responses of basketball players. International Journal of Environmental Research and Public Health, 19(8), 4810. https://doi.org/10.3390/ijerph19084810
Specific comments.
Title. The title of the manuscript is correct.
Abstract. Incorporate in the summary, a more precise sentence of the results.
Introduction. This section presents the problem in a coherent and clear manner with the correct support of the scientific literature. However, it is convenient to update the references, since there are different documents related to the subject and no mention is made, and it would even be interesting to mention the different existing studies related to Basketball and Music. Also, it could be a future study of review. Some bibliographical references are attached to carry out the section of Basketball and Music:
Blasco-Lafarga, C., Ricart, B., Cordellat, A., Roldán, A., Navarro-Roncal, C., & Monteagudo, P. (2022). High versus low motivating music on intermittent fitness and agility in young well-trained basketball players. International Journal of Sport and Exercise Psychology, 20(3), 777-793.
Boolani, A., Lackman, J., Baghurst, T., LaRue, J. L., & Smith, M. L. (2019). Impact of Positive and Negative Motivation and Music on Jump Shot Efficiency among NAIA Division I College Basketball Players. International Journal of Exercise Science, 12(5), 100-110.
Hammami, R., Nebigh, A., Selmi, M. A., Rebai, H., Versic, S., Drid, P., ... & Sekulic, D. (2021). Acute Effects of Verbal Encouragement and Listening to Preferred Music on Maximal Repeated Change-of-Direction Performance in Adolescent Elite Basketball Players—Preliminary Report. Applied Sciences, 11(18), 8625. https://doi.org/10.3390/app11188625
Methods. Add the Design section.
- Study design. To write the design section, we recommend that you take some of the following methodologists as references.
Ato, M., López-García, J.J., & Benavente, A. (2013). A classification system for research designs in psychology. Annals of Psychology, 29(3), 1038-1059.
Results. Summary of study data and table are correct.
Discussion. The section Discussion is correct.
Conclusion. Differentiate the discussion of the main conclusions of the study. To do this, you must create this section. And modify the limitations of the study and locate them in said section at the end. Also, they must be direct, and highlight the main contributions of the study.
References. They should be reviewed and updated according to the publication standards. There are many errors in the references. Therefore, correct them and adapt them to the magazine's regulations.
Author Response
Se file.
